# Differential Regulation of Type I and Type III Interferon Signaling

**DOI:** 10.3390/ijms20061445

**Published:** 2019-03-21

**Authors:** Megan L. Stanifer, Kalliopi Pervolaraki, Steeve Boulant

**Affiliations:** 1Schaller Research Group at CellNetworks, Department of Infectious Diseases, Heidelberg University Hospital, 69120 Heidelberg, Germany; k.pervolaraki@dkfz-heidelberg.de; 2Research Group “Cellular Polarity and Viral Infection” (F140), German Cancer Research Center (DKFZ), 69120 Heidelberg, Germany

**Keywords:** Interferon, type I IFN, type III IFN, interferon signaling, JAK-STAT, signal transduction

## Abstract

Interferons (IFNs) are very powerful cytokines, which play a key role in combatting pathogen infections by controlling inflammation and immune response by directly inducing anti-pathogen molecular countermeasures. There are three classes of IFNs: type I, type II and type III. While type II IFN is specific for immune cells, type I and III IFNs are expressed by both immune and tissue specific cells. Unlike type I IFNs, type III IFNs have a unique tropism where their signaling and functions are mostly restricted to epithelial cells. As such, this class of IFN has recently emerged as a key player in mucosal immunity. Since the discovery of type III IFNs, the last 15 years of research in the IFN field has focused on understanding whether the induction, the signaling and the function of these powerful cytokines are regulated differently compared to type I IFN-mediated immune response. This review will cover the current state of the knowledge of the similarities and differences in the signaling pathways emanating from type I and type III IFN stimulation.

## 1. Introduction

Six years ago Isaacs and Lindemann made an amazing discovery when they described molecules able to “interfere” with influenza virus replication [1]. These molecules are now referred to as interferons (IFNs) and can protect humans and animals against a broad range of viruses and pathogens by generating IFN-mediated innate immune responses [2]. The first interferon described was type I IFN [1]. In humans, there are 17 subtypes of type I IFNs including 13 subtypes of IFN-α, along with IFN-β, IFN-ε, IFN-κ, and IFN-ω [3,4,5,6]. Type I IFNs have a broad range of biological functions including modulation of innate and adaptive immune responses, anti-proliferative functions, and most importantly antiviral activities [5,7]. Remarkably, although all type I IFN subtypes share limited structural similarity, they all share the same heterodimeric receptor complex constituted of one chain of the IFN-α receptor 1 (IFNAR1) and one chain of the IFN-α receptor 2 (IFNAR2) [8].

In contrast with the type I IFN family, type II IFN family consists only of IFN-γ, which has a very low sequence similarity with the other IFNs [9]. While IFN-γ is mainly produced by immune cells (activated T cells, natural killer cells and macrophages) and is described as an immunomodulatory cytokine, it also shows a response against viruses, bacteria and parasites [10]. Contrary to the other IFNs, IFN-γ signals as a homodimer using the IFN-γ receptor (IFNGR) complex. The IFNGR ternary complex consists of two chains of IFN-γ receptor 1 (IFNGR1) and two chains of IFN-γ receptor 2 (IFNGR2) and displays a broad tissue expression pattern [11].

Beside type I and type II IFNs, another family of IFN was recently described by two independent research groups and referred to as type III IFNs or IFN-λs [12,13]. In Homo sapiens, this family contains four described members including IFN-λ1 (IL-29), IFN-λ2 (IL-28A), IFN-λ3 (IL-28B), and most recently, IFN-λ4 [12,13,14]. The heterodimeric receptor used by type III IFNs to signal is different from the receptor used by type I IFN and consist of the IFNLR1 chain (a.k.a. IL-28Rα) and the IL-10R2 chain. Importantly, the IL-10R2 chain is also part of the heterodimeric receptor of all the IL-10 family (IL-10, IL-22 and IL-26) [3,12,13]. Expression of IFNLR1 is not ubiquitous but appears limited to cells of epithelial origins (e.g., hepatocytes, intestinal cells and lung) and to a specific subset of immune cells (i.e., NK cells, pDCs, and DCs) [15,16,17,18,19,20,21,22]. This limited tropism of the type III IFN receptor has led over the years to the model where type III IFNs have unique properties at mucosal surfaces [16,21,22,23,24,25,26,27,28,29].

Although type I and III IFNs are non-related from both sequence and structure perspective as well as using two different receptor complexes for their signaling, both cytokines induce a remarkably similar panel of interferon stimulated genes (ISGs). This observation led to the hypothesis that both cytokines were functionally redundant [29,30,31,32,33,34]. However, in the last few years, growing evidence highlights significant differences in the mode of action and functions of type I vs. type III IFNs [21,22,25,27,35,36]. Apart from the differences in the transcriptional regulation of ISGs after type I versus III IFN stimulation, which has been detailed in previous reviews [37,38,39], crucial differences have begun to be unraveled regarding the signaling pathways emanating from the receptor complexes. It has been shown that there are different levels of both the type I and type III IFN receptors at the plasma membrane and different binding affinities for each IFN [40,41,42,43,44]. Additionally, the activation and regulation of the downstream signaling components and their ability to drive ISG production is distinct to each IFN providing type I and III IFNs with unique functions. This review will focus on the differences emanating from receptor activation to the regulations of ISG induction following either type I or III IFN stimulation.

## 2. Canonical Type I and Type III IFN Signaling

Both type I and type III IFNs can promote signaling in an autocrine (on the secreting cells) or paracrine (on bystander cells) manner to induce an IFN-mediated immune response. Binding of IFNs to their respective receptors induces the formation of a receptor ternary complex mediating signal transduction. First, binding of type I and III IFNs to the extracellular part of the IFNAR1/2 and IFNLR1/IL10R2, respectively, induces conformational changes in the intracellular part of the receptor subunits. Both heterodimeric receptors do not display any catalytic activity, instead, members of the receptor-associated Janus kinases (JAK) family (JAK1, JAK2 and TYK2), which are constitutively associated with the heterodimeric receptors, will be activated upon binding of IFNs. In turn, they subsequently mediate the phosphorylation of tyrosine residues on the intracellular domains of the IFN receptors. This activation allows for the recruitment of signal transducer and activator of transcription (STAT) proteins, which are then phosphorylated by JAKs. Phosphorylation of STAT induces their dimerization and the assembly of the IFN-stimulated gene factor 3 (ISGF3) complex, consisting of STAT1, STAT2, and IRF9. ISGF3 and dimers of STATs act as transcription factors and upon translocation to the nucleus regulating ISG expression (Figure 1). Regulation of this IFN-mediated signaling can occur at multiple levels within the signal transduction pathways by 1) receptor internalization 2) regulation of JAK/STAT activation 3) activation of JAK/STAT independent signaling pathways and 4) regulation of promoter elements.

All the IFN receptors belong to the class II helical cytokine receptors (hCRs) family. Class II receptors function as heterodimers and bind monomer or homodimers of ligands consisting of six α-helices [45]. Additionally, class II receptors are characterized by a transmembrane domain which lacks the WSXWS amino acid sequence of class I helical receptors [3,46]. For the type I IFN heterodimeric receptor, IFNAR2 displays a high affinity for IFN binding [41,42,47,48,49]. Interestingly, although IFNAR1 has a low affinity for IFN, it is responsible for the distinction of the different subtypes of type I IFN [50,51]. It has been reported that the binding affinities between type I IFNs and IFNAR1/2 dramatically varies and correlates with their differences in biological activity and different sensitivities to negative regulators [42,43]. While all IFN-α’s present a similar low affinity for IFNAR1 and are highly sensitive to the negative regulator USP18, IFN-β binds IFNAR1 more tightly and its activity seems to be unaffected by USP18 (Figure 2) [44,52]. 

Similar to the type I IFN receptor, the type III IFN receptor also displays different affinities for IFN in a chain-specific manner however, the overall binding affinity for type III IFNs is much lower compared to type I IFNs [53,54]. The IFNLR1 chain corresponds to the high binding affinity subunit, while the IL-10R2 displays a low affinity for type III IFNs (Figure 2) [46,47]. Unlike IFNAR1 and IFNAR2 which are strictly involved in mediating type I IFN signaling, the IL-10R2 chain of the heterodimeric type III IFN receptor is also involved in mediating IL-10, IL-22 and IL-26 signaling. Importantly, this shared receptor subunit has been described to mediate a potential crosstalk between IL-22, IL-10 and type III IFNs. Type III IFN activity is inhibited in the presence of IL-10 and it has been suggested that this is due to a competition of type III IFNs with IL-10 for the IL-10R2 receptor [55]. Interestingly, IL-22 was shown to act in a synergistic manner with type III IFNs leading to an increase in STAT1 activation which then leads to an increase in the type III IFN dependent ISG expression [24]. It is unclear how interactions with one IL-10 family member lead to a decrease in IFN activity while the other member increases its activity. However, as type III IFNs themselves induce the production of IL-10 (as an ISG) this may serve as a negative feedback to control its activity. On the contrary, IL-22 will serve as a positive feedback loop on type III IFN-mediated signaling in the intestinal mucosa. In this environment, intestinal epithelial cells, in response to pathogen challenges, will secrete IL-1α which in turn will activate intestinal specific innate lymphoid immune cells (ILC3) promoting the secretion of IL-22. Similarly, DCs sensing the lumenal content of the gut will secrete IL-23 which in turn will activate ILC3 again promoting IL-22 secretion to signal on intestinal epithelial cells. As such, the IL-22 feedback loop can be seen as a strategy to further boost the antipathogen protection at times of infection and tissue damages and local inflammation [24,56].

In general the regulation of receptor cell surface levels by internalization and degradation, is considered to be the most specific and rapid cellular strategy to regulate and to limit their signaling [57]. After IFN binding, the ternary IFNAR1/2 complex is internalized by clathrin-mediated endocytosis [58,59]. Interestingly while IFNAR1 is rapidly routed for lysosomal degradation [60,61], IFNAR2 can be either recycled back to the cell surface or degraded [62,63]. In particular, a linear tyrosine based sequence in IFNAR1 serves as an endocytic motif, which is recognized by the clathrin-adapter protein AP-2 endocytic complex. Upon recognition of this motif AP-2 is recruited to the plasma membrane and begins to further recruit the clathrin-machinery [58,59]. This leads to the formation of the clathrin-coated vesicles and internalization of the ternary IFNAR complex. In basal conditions, this tyrosine motif is constitutively associated with TYK2 and remains masked from AP-2 [58,59]. Upon stimulation, phosphorylation of this motif decreases its binding with TYK2 and facilitates its exposure to AP-2 explaining the importance of TYK2 for IFNAR1 cell surface stability [64]. As such, the fate of type I IFN signaling is tightly linked to the internalization and degradation of the ternary IFNAR1/2 complex, which is regulated by tyrosine endocytic sorting motifs in IFNAR1. Importantly, IFNAR1 levels can also be controlled in a ligand independent manner through the viral induced activation of the unfolded protein response leading to the subsequent ubiquitination, internalization, and down regulation of the IFNAR1 signal (Figure 2) [60,65]. 

To date, there is no literature available concerning the molecular mechanisms underpinning the IFNLR1/IL-10R2 ternary complex internalization and recycling. However, the basal levels of its most closely related family member, IL-22R, displays a similar restricted expression in epithelial cells [66,67] and its expression is regulated through ubiquitination, leading to a fast turnover of the receptor [68]. Importantly, phosphorylation by GSK-3β leads to a stabilization of the receptor and modulates its activity. Comparably, the IL-10R2 receptor levels are also modulated through ubiquitination leading to its internalization and degradation [69]. Similar to IFNAR1 expression IL-10R2 levels can also be modulated in a ligand-independent manner through pattern recognition sensing leadings to its downregulation [70]. Interestingly, while its ligand dependent internalization has not been characterized, it has been shown that IFNLR1 surface levels can be modulated by virus infection, where rotavirus leads to the down regulation of both receptors in a lysosomal dependent manner [71].

## 3. Regulation of JAK/STAT Activation

### 3.1. JAK Activation/Inactivation

Besides the direct regulation of receptor levels, one of the primary mechanisms by which cells can regulate IFN-mediated signaling is through regulation of IFNAR1/2 and IFNLR1/IL-10R2 interactions with downstream signaling molecules (e.g., JAK family members). TYK2 and JAK1 are constitutively bound to IFNAR1 and IFNAR2, respectively. Upon type I IFN binding, structural rearrangements of the two subunits of the type I IFN receptor induce the juxtaposition of the two kinases located in TYK2 and JAK1. This results to their auto phosphorylation and their transphosphorylation of the amino acid Y1022 and Y1023 on JAK1 and Y1054 and Y1055 on TYK2 [72,73]. Activated JAKs, in turn, mediate the phosphorylation of both IFN receptor chains creating multiple binding sites for both STAT1 and STAT2 through their SH2 domain. While the functions of all phosphorylation sites remain elusive, amino acids Y337 and Y512 are known to be important for STAT1 and STAT2 recruitment [74], while the IFNAR1 tyrosine Y466 has been shown to be important for STAT2 activation [75]. Additionally, residues 525 to 544 of IFNAR1 have been shown to be critical for its regulation. IFNAR1 constructs lacking amino acids 525-544 show an increased sensitivity to IFN suggesting that theses residues could be sites where negative regulators bind to control its activity [76]. In addition, in mouse cells tyrosine residues Y510 and Y335 on IFNAR2 were shown to be critical for IFN dependent ISG induction as their absence lead to a reduced ability to activate STAT1 and possibly STAT2 [77].

Similar to the type I IFN pathway, TYK2 and JAK1 have been found to be associated with IL-10R2 and IFNLR1, respectively. While it is clear that type III IFNs require JAK1 for their downstream signaling pathways, they are able to signal in the absence of TYK2 [78]. This ability of type III IFN to signal in the absence of TYK2 was first recognized by evaluating multiple patients who displayed TYK2 deficiencies. These patients often showed signs of skin disorders and bacterial infections but only displayed a mild susceptibility to viral infection [79]. Further investigation showed that the lack of TYK2 impacted the ability of type I IFNs to clear viral infection while type III IFNs maintained their antiviral capacities showing that TYK2 is dispensable for the type III IFN mediation signaling of IL-10R2. While TYK2 is clearly associated with the IL-10R2 receptor but appears to be not important for type III IFN mediated signaling, TYK2 plays a key role for the signaling of other IL-10 family members such as IL-10 itself [78]. These observations are interesting and raise further questions about JAK signaling downstream type III IFN receptor activation. As it has been shown that JAK1 and TYK2 trans-activate each other upon type I IFN binding of IFNAR1/2, it will be interesting to understand, through further studies, whether the loss of this trans-activation by JAK1 leads to the differences seen in the downstream signaling following type III IFN stimulation. Although the IFNLR1 phosphorylation has not been investigated in as much detail as the IFNAR1/2 receptors, two amino acids on IFNLR1 (the tyrosine Y343 and Y517) are critical for STAT2 activation. Particularly, the motif surrounding the Y343 in IFNLR1 strongly resembles the motif surrounding the Y466 of IFNAR1. Similarly, the sequence surrounding the Y517 of IFNLR1 displays some degree of resemblance with the C-terminal part of IFNAR2 containing the Y512 [80,81]. Importantly, although these motifs reflect some degree of similarity in the STAT2 docking sites between the IFNLR and the IFNAR complexes, their intracellular regions are dramatically different. Due to these differences it is not surprising that differences in the TYK2 requirements and in the kinetics of STAT activation by type I and III IFN mediated signaling have been reported [31,78].

As the signal transduction downstream of both type I and type III IFN receptors is mostly based on phosphorylation, several phosphatases have been described to modulate IFN-mediated signaling. Specifically, the protein-tyrosine phosphatase 1B (PTPB1) has been shown to bind directly to the IFNAR1 subunit and modulate its AP-2 dependent endocytosis by removing the phosphate on Y466 thereby activating its uptake [82]. This regulation has only been shown in human cells where TYK2 binding modulates Y466 phosphorylation thereby maintaining basal levels of IFNAR1 at the cell surface. Mouse cells do not require TYK2 for the stable surface expression of IFNAR1 and are therefore insensitive to the presence of PTPB1 [82]. TYK2 and JAK2 can also be modulated by PTPB1. In the absence of PTPB1, JAK2 becomes hyperphosphorylated leading to excessive IFN signaling (Figure 2) [83]. Interestingly, another related phosphatases T cell protein-tyrosine phosphatase (TCPTP) shows a similar ability to modulate JAK2 activity and mice lacking either PTPB1 or TCPTP display similar over activations of IFN-mediated signaling [84]. Additionally, the transmembrane PTPase CD45 has also been shown to serve as a JAK phosphatase in a type I IFN dependent manner [85]. The expression and role of these phosphatases have mostly been studied in hematopoietic cells where type I and II IFNs play a key role for signaling but type III IFN receptors are not expressed. Additionally, these phosphatases have been shown to bind members of the JAK family, however what activates their recruitment to bind JAKs is still unclear. Further work is needed to determine whether these phosphatases will also regulate type III IFN mediated signaling and what signals are needed to drive their negative regulation of type I mediated signaling. 

While the regulation of phosphorylation plays a key role in JAK activation other regulators, which compete for the binding of JAK to the IFNAR1/2 complex, have been identified. The ubiquitin-specific protease USP18, binds to IFNAR2 and prevents its interaction with JAK1 (Figure 2) [86]. This association seems to be ligand dependent, as it inhibits specific signaling from IFN-α subtypes, which bind IFNAR2 with lower affinity than IFN-β [44,52,87]. USP18 dependent inhibition of IFN-α responses, could be an explanation for the shorter duration of IFN-α responses compared to IFN-β [32,33]. High USP18 levels are also suggested to be the reason that many hepatitis C patients show a refractory phenotype to IFNα based antiviral therapy [52]. USP18 does not play a role in regulating the response by type III IFNs under normal conditions as the type III IFN receptor complex lacks consensus sites for USP18 binding [87]. Although, a recent study showed that in a mouse mammary tumor model, the loss of USP18 lead to an increase in type III IFN signaling which lead to a lower tumor burden. This suggests that USP18 can act on type III IFN signaling, however this was in a tumor model and it is unclear if there whether other parts of the signaling cascade were disrupted in this model or if USP18 was expressed at higher than normal levels [88]. Additionally, a possible explanation for the USP18 dependent regulation of IFNLR1 could be due to a crosstalk occurring between the IFNAR1 and IFNLR1 receptors, where USP18 could directly affect IFNAR1 and this could lead to changes in IFNLR1 signaling.

In addition to competing for binding of the JAK family members to the IFN receptors, other negative regulators are known to directly interact with these JAK proteins to negatively regulate their activation. The suppressor of cytokine signaling (SOCS) (e.g., SOCS1 and SOCS3) are considered the most potent negative regulators as they can directly interact with TYK2 interfering with its activation [89,90]. SOCS1 knock-out mice show uncontrolled IFN signaling leading to an inflammatory state. SOCS1 has been shown to exert its effect specifically on the IFNAR1 receptor while IFNAR2 regulation is independent of SOCS1. SOCS1 does not bind to the IFNAR1 receptor directly but modulates it activity through TYK2 (Figure 2). SOCS1 binds to TYK2 leading to its ubiquitination-dependent destabilization, which subsequently drives IFNAR1 downregulation [90,91]. Additionally, unlike USP18, SOCS1 regulates type III IFN mediated signaling. Overexpression of SOCS1 in hepatic cells lines has been shown to reduce the type III IFN mediated induction of p-STAT1 and ISGs as well as interferon sensitive response elements (ISRE)-activation [92,93]. Importantly, these in vitro observations were confirmed in vivo where SOCS1 knock-out mice also show increased ISG induction in response to type III IFNs [87]. These results are interesting given the observation that TYK2 is not necessary for type III IFN signaling and SOCS1 typically acts through regulation of TYK2. While the observations that SOCS1 regulates type III IFN signaling were mainly performed in mouse hepatocytes, the TYK2-independent type III IFN signaling observations were mainly made in human cells. This may speak for tissues and/or species specific differences in the regulation of the downstream IFN signaling pathways.

Importantly, as USP18 and SOCS are ISGs, the timing of their induction may determine the negative feedback loop seen in both type I and III IFN signaling [93]. A sustained upregulation of SOCS1 by type III IFNs has been shown in hepatocytes and is proposed to be responsible for the long lasting regulation of SOCS1 on IFNλ1 mediated response [87], while IFNα causes a rapid and transient induction of SOCS1 which may be responsible for the early inhibition of IFNα signaling and the short duration of IFNα mediated ISG responses [87,91,94]. Similarly, in human intestinal epithelial cells, it was reported that SOCS1 is induced early upon addition of type I IFN (3-6h) while type III requires 24 h for its induction [29] further supporting the idea of a quick regulation of type I IFN signaling and a long lasting activity of type III IFN signaling due to a delay in the transcription of the negative regulator SOCS1. Additionally, it has been shown that in murine hepatocytes, USP18 is induced late after IFN stimulation and its protein levels increased over time, correlating with the long lasting refractoriness of hepatocytes to IFNα signaling [44,87,94]. A significant delay in the kinetics of USP18 expression has also been observed in human intestinal epithelial cells treated with type I and type III IFNs, respectively [29]. However, further investigation is required to determine whether there is a direct correlation between the kinetics of induction of USP18 (and SOCS1) with the kinetics of ISGs induction in type I versus type III IFN treatment in human cell lines.

### 3.2. Regulation of STATs

In the canonical type I and type III IFN signaling pathway, the proteins responsible for signal transduction downstream both type I and III IFN receptors and JAKs are STAT1, STAT2 and IRF9. As these proteins are interferon-stimulated gene themselves, transcriptional regulation of their expression represents an important feedback loop regulating IFN signaling. For example, it has been shown that multiple cell types secrete low levels of IFN-β under homeostatic conditions which mediates basal levels of STAT1, STAT2 and IRF9 [95]. The origin of this basal secretion of type I IFN is not fully understood but, in the gut, it has been proposed that under homeostatic conditions, both epithelial and immune cells can sense the commensal flora leading to a controlled production of low levels of IFNs [96]. Contrary to these homeostatic conditions where low levels of IFNs are secreted by cells, the presence of pro-inflammatory signals such as TNF-α and IL-6 can lead to higher expression levels of STAT1 and IRF9 which ultimately result in a prolonged ISG expression and sustained antiviral protection [97,98,99]. For type I IFNs, in addition to STAT1/2, STAT3-6 have been shown to be transcriptionally upregulated, in turn enhancing the antiviral and anti-proliferative actions of these IFNs [100,101,102,103]. While STAT1, STAT2 and STAT3 are induced downstream type I IFNs in most all cell types, STAT4, STAT5 and STAT6 are induced in a cell type dependent manner and results in unique signaling complexes in these cell types (e.g., STAT5 activation along with the CrkL adapter lead to the induction of a specific subset of ISGs) [104]. Similar to type I IFNs, STATs 1-5 have been described to be induced downstream the type III IFN receptor [80,92,105]. However, whether activation of STAT 3, 4, 5 following type III stimulation is associated with cell type specific functions and gene inductions remains to be fully addressed (Figure 3).

Beside the canonical ISGF3 transcriptional complex made of STAT1:STAT2 heterodimer and of IRF9, almost all combination of STAT homo- and heterodimers can be found: STAT1:STAT1, STAT3:STAT3, STAT4:STAT4, STAT5:STAT5, STAT6:STAT6, STAT1:STAT2, STAT1:STAT3, STAT1:STAT4, STAT1:STAT5, STAT2:STAT3 STAT5:STAT6 [12,30,92,100,106,107]. ISGF3 binds a specific sequence denoted as the IFN sensitive response element (ISRE) in the promoter region of ISGs. The STAT homo- and heterodimers show a binding affinity for the IFN-γ activated site (GAS) element within the promoters of ISGs. Importantly, in the promoter region of ISGs multiple combinations of STAT binding elements can be found such as ISRE’s alone, GAS elements alone or both ISRE and GAS elements together (Figure 3). This abundance of transcription factors and multiplicity of binding sites within the promoter regions of ISGs is likely to allow for a fine regulation of gene expression and to permit the expression of different subsets of ISGs by differential induction of STAT-containing complexes [100,106,107,108].

Phosphorylation of STAT1 (Y701) and STAT2 (Y609) by JAK leads to their dimerization and nuclear translocation. In addition to these tyrosine residues, numerous additional modifications have been shown to also positively regulate STAT activation. Serine S727 phosphorylation of STAT1 and STAT3 by specific kinases upon type I IFN stimulation have been shown to enhance their transcriptional activity without being required for their translocation to the nucleus [109,110]. Protein kinase C (PKC) family members [32,111,112], cyclin-dependent kinase 8 [113,114] and p38-MAPKinases [115] have been also reported to mediate S727 phosphorylation following type I IFN treatment. Additionally, S708 phosphorylation of STAT1 by NF-kB kinase-e (IKKe) was shown to enhance antiviral activity through promoting ISGF3 formation and transcriptional activity enhancement [116,117]. Similarly, a study by Bolen and co-workers, has observed phosphorylation of S727 on STAT1 following stimulation of IFNλ1, λ2 and λ3 in hepatocytes suggesting that it will also act in a positive manner for type III IFN mediated signaling however this enhanced transcriptional activity has not yet been directly tested [32].

In parallel to these mechanisms of positive regulation of IFN signaling, numerous negative regulators interfering with STATs activation and functions have been identified. The main negative regulators are members of the protein tyrosine phosphatase (PTP) family [118]. For example, upon type I IFN stimulation, the Src Homology phosphatase 2 (SHP-2) (constitutively associated with IFNAR2) becomes activated and tunes down the signaling cascade by interfering with the phosphorylation of STAT1, STAT2 and JAK1 [82,119,120]. Apart from the phosphatase-dependent STAT deactivation, phosphorylation at non-canonical residues (S287 and T387 on STAT2) has been linked to the negative regulation of the type I IFN-mediated transcriptional, antiviral and anti-proliferative actions [121,122]. Protein inhibitors of activated STAT (PIAS) family members can also negatively regulate STAT1 by binding this transcription factor and as a consequence blocking its transcriptional activity. In addition, STAT1 is also sumoylated by PIAS1 [123,124], but the functional importance of this modification for IFNAR-induced gene activation needs to be further elucidated. Additionally, acetylation and methylation sites on STATs have been reported and are proposed as sites for further important post-translational modifications in their regulated architecture [125,126,127]. Whether these negative modifications are induced upon type III IFN stimulation has not yet been addressed. However, it will be interesting for future studies to compare the acetylation, methylation and sumoylation pattern of STATs downstream IFNAR1/2 and IFNLR1/IL-10R2 activation and determine their contribution to ISG induction.

In addition to the formation of the canonical ISGF3 complex, in certain conditions the formation of alternative non-canonical ISGF3 complex containing unphosphorylated STATs has been reported [128,129]. As STAT1, STAT2 and IRF9 are ISGs, they are produced in large amounts following stimulation by both type I and III IFNs. These unphosphorylated components form the U-ISGF3. This U-ISGF3 binds to unique ISREs and produces a set of around 30 ISGs [130]. These ISGs are able to remain in the cell for days after the initial IFN stimulation, long after p-STAT1 and p-STAT2 have returned to basal levels. It has been suggested that the canonical ISGF3 is responsible for the rapid response to IFNs while the U-ISGF3 persists and keeps the cells protected for days [128,130]. While most of the ISGs produced by the U-ISGF3 are antiviral there are a few key genes produced which protect the cells from DNA damage [130]. In addition, other non-canonical forms of ISGF3 have been reported: the ISGF3^II^ complex including phosphorylated STAT1, unphosporylated STAT2 and IRF9, and the ISGF3-like complex formed by STAT2 and IRF9 have been shown to be assembled mainly upon type I and II IFN treatment leading to prolonged ISG induction [117,131,132,133,134]. 

## 4. Alternative Signaling Pathways Downstream Type I and III IFNs

Apart from the classical JAK-STAT signaling axis, there is growing evidence linking IFN stimulation to a JAK-STAT independent transcription of ISGs through pathways such as the Crk-like protein (CrkL)–Ras related protein 1 (RAP1) pathway [106,107,135,136,137,138], the phosphatidyl-inositol 3-kinase (PI3K)-signaling pathway [139] and the mitogen-activated protein kinase (MAPK) pathway [30,140,141,142]. 

### 4.1. CrkL-RAP1 Pathway in Interferon Signaling

The major role of type I IFNs is the production of an antiviral state within cells. However, they also play a role in the proliferation of both hematopoietic and non-hematopoietic cells through the activation of the CrkL [106]. It has been shown that STAT5 is constitutively associated with the IFNAR1/2 complex through its interaction with TYK2. Upon Tyk2 phosphorylation, STAT5 becomes phosphorylated which produces a docking site for CrkL. This forms a complex of STAT5:CrkL which translocates into the nucleus and binds to GAS elements driving the production of growth inhibitory genes (Figure 3) [137,138]. Additionally, independent of STAT, CrkL phosphorylation by IFNAR bound CBL has been shown to promotes its association with C3G, the guanine exchange factor for Rap-1. This interaction leads to the activation of Rap-1 further promoting its tumor suppressor activity [143]. Multiple type I IFNs have been shown to activate CrkL suggesting that it is a universal mechanism leading to changes in the growth properties of the cell. So far there have not been any reports linking type III IFN stimulation with CrkL activation.

### 4.2. Phosphatidyl-Inositol 3-Kinase (PI3K)-Signaling Pathway

The PI3K pathway has been shown to have a dual role in IFN signaling. It not only acts upon the production of antiviral effectors but also regulates the transcription and translation of ISGs [106]. The best characterized IFN-dependent PI3K signaling cascade initiates through the activation of the insulin receptor substrate (IRS)-1 [144]. The binding of type I IFNs, IFNα, IFNβ and IFNω, leads to the rapid but transient phosphorylation of IRS-1. Activated IRS-1 is capable of binding to the catalytic p85 subunit of PI3K which subsequently leads to the activation of the regulatory p110 subunit of PI3K [145]. Importantly cells which lack the catalytic subunit of PI3K are deficient in the production of a subset of ISGs, Akt-dependent mTOR activation and mRNA translation, demonstrating the crucial role of PI3K in multiple IFN-dependent signaling pathways [146]. Upon type III IFN treatment, phosphorylation of Akt has been shown, which was blocked with the use of a PI3K inhibitor [92]. However further studies are needed to validate the involvement of the PI3K pathway in type III IFN signaling. The inter connections of each of these pathways and their effects on the cells are still under debate.

### 4.3. MAP Kinases in Type I and Type III IFN-Mediated Signaling

The extracellular signal-regulated kinase (ERK), the c-Jun N-terminal kinase (JNK) and the p38 MAP kinases have been shown to be activated by type I IFN treatment in multiple cell lines [30,140,141,142]. p38 kinase has been the most thoroughly characterized and was shown to be critical for the antiviral functions of type I IFN in specific cell lines and *in vivo* mouse models [140,147,148,149,150]. Within the signal transduction pathway downstream type I IFN, the RAC1 guanine-nucleotide-exchange (GEF) factor VAV gets activated by JAKs [151,152,153]. After the VAV-dependent RAC1 induction, RAC1 mediates the activation of the MAP kinase kinase 3 (MKK3) and MAP kinase kinase 6 (MKK6) [154]. Subsequently, these kinases phosphorylate and activate p38. Multiple downstream targets of p38 have been shown to be activated. For example, the mitogen- and stress-activated kinase 1 (MSK1) and MSK2, the MAPK-activated protein kinase 2 (MAPKAPK2), MAPKAPK3 and MAPK-interacting protein kinase 1 (MNK1) which in turn appear to play an important role in establishing and/or regulating the type I IFN responses [107]. Contrary to the p38 signaling pathway, the type I IFN dependent activation of ERK and JNK has not been investigated thoroughly. Only fragmented information is available where some reports describe a cell specific activation of ERK upon IFN-α stimulation [155,156] and even less information is available concerning the involvement of JNK during type I IFN-mediated antiviral response [157].

Analogous to type I IFNs, type III IFN treatment of cells has been reported to activate MAP kinases in multiple cell types [28,30,92,158]. Interestingly, it was shown that the response of epidermal fibroblasts to type III IFN is MAP kinase dependent and leads to ISG induction and TGF-β induced collagen production. This recent study is an indication for a specific role of IFNLR-dependent MAP kinases activation in antiviral protection and repair processes of epidermal tissue [158]. Additionally, it has been recently shown that MAPKs are important for type III but not type I IFN mediated antiviral protection in human intestinal epithelial cells (IECs) [28]. While both type I and III IFNs were able to activate p38, ERK and JNK in IECs, inhibition of their activation only affected the ability of type III IFNs to protect against viral infection. Together these data strongly suggest that type I and III IFNs have different downstream signaling pathways leading to their antiviral state.

## 5. Modulation of ISG Expression

### 5.1. Role of Additional Transcription Factors in ISG Expression

Members of the interferon regulatory factor (IRF) family (e.g., IRF-1) are transcriptionally upregulated upon IFN stimulation of cells. These transcription factors, in turn induce the transcription of a subset of ISGs by binding to specific promoter regions (IRF-E sites) which overlap with the ISRE sequence [159,160,161,162]. It is believed that the combinatorial action of both ISGF3 and IRFs on the promoter regions of ISGs can improve/regulate their transcription. On the contrary, it was shown that by competing for the same DNA binding sites, IRF-2 competes with IRF-1 and IRF-9 and act as a suppressor of type I IFN response [163,164,165,166]. In addition, it was shown that expression of certain ISGs (e.g., MxA, GBP, MHC class I and B2M in immune cells) can be activated directly by IRFs in a STAT independent manner [167] [168]. Furthermore, upon IFN or virus stimulation, IRF-7 expression amplifies various ISGs including IFN-α itself, which further enhances IFN responses through a positive feedback loop [169,170]. Type III IFN has been shown to induce IRF-7 in a similar manner to type I IFNs suggesting some overlap in the amplification of IFN signaling [30]. In addition, IRF-1 has been shown to be required for the production of type III IFN downstream peroxisomal MAVS stimulation following viral infection suggesting a critical role for this factor in the production and amplification of type III IFN response [171]. Interestingly, IRF-1 has been recently shown in human respiratory cells to be uniquely induced by type I IFN [172]. Type III IFN induces very low or weak levels of IRF-1 transcript and protein in human respiratory cells further showing that each interferon drives a unique antiviral program [173]. 

### 5.2. Epigenetic Regulation of IFN and ISGs

Epigenetic regulation of gene expression has been proposed as an additional mechanism, which helps to regulate cell type specific gene profiles driving unique IFN transcript and ISG profiles. Currently there are more than a dozen post-translational modifications that have been shown to be present on the tails of histones, which include methylation, acetylation, ubiquitination, sumoylation and phosphorylation [174]. Except for methylation, all other modifications have been shown to lead to loosening of the histone/DNA complex due to changes in the charge on the histone molecules resulting in gene expression activation [174]. Methylation can lead to both activation and repression of transcription depending on whether the methylation is mono-, di or tri. Di-methylated histone 3 lysine 9 (H3K9me2) has been associated with transcriptional repression [174]. Interestingly, professional immune cells (i.e., DCs) in both mice and humans show decreased levels of the H3K9me2 repression markers at both the IFNβ gene as well as ISGs such as Mx1 and IFIT1 in comparison to “non-professional” immune sensing cells (i.e., mouse embryonic fibroblasts). It was shown that this difference was due to the differential level of G9a expression, a lysine methyl transferase, in both DCs and fibroblast. When MEFs were depleted of G9a function they lost this repressive mark at the IFN and ISG promoters and became capable of inducing much higher IFN and ISG levels in response to pathogen stimuli. Interestingly, the activation marker H3K4me2 is also highly present at both IFN and ISG promoters in DCs. This is characteristic of poised transcription, allowing these specific genes to be ready for quick transcription following release of the “poised program” [175]. Additionally, many viruses target epigenetic markers and modify them during their infection to subvert the IFN induced antiviral program of infected cells (e.g., Influenza has been shown to change total DNA methylation levels and to down regulate histone deacetylate activity) [176]. 

The role of epigenetic changes downstream of type III IFN is currently less well understood. It was recently shown in mouse intestinal epithelial cells, that the histone deacetylase (HDAC) activity controlled the amount of type III IFN responsive cells within a population. In this study, Bhushal et al., showed that under normal conditions type III IFN was only able to induce ISG expression in a subset of stimulated intestinal epithelial cells. This restriction was seen even at high doses of type III IFN. Interestingly, when HDAC inhibitors were added to the cells, type III IFN was capable of inducing ISG levels in a larger percentage of intestinal epithelial cells. This phenotype was unique to type III IFN signaling as type I IFN was insensitive to HDAC inhibitor treatment suggesting that type I and III IFN have different requirements for chromatin activation and ISG induction [177]. Additionally, the IFNLR1 receptor itself has been shown to be modulated by HDACs. It was shown that the addition of HDAC inhibitors lead to the induction of IFNLR1 gene levels which correlated with a gain of function in both antiviral and anti-proliferative properties in previously non-responsive cells [178]. 

Apart from histone markers, variants of histones exist within cells. The best-characterized variant is H3.3, which is known to contribute to epigenetic memory during development [179]. Interestingly, type I IFN treatment of cells has been shown to lead to the accumulation of H3.3 in the promoters of ISGs [180]. This incorporation then leads to a transcriptional memory affect, which allows for greater p-STAT1 and Pol II recruitment during subsequent IFN treatments [181]. Whether H3.3 is also induced under type III IFN treatment has currently not been addressed.

In addition to histone modifications, many proteins act to modulate chromatin through remodeling. The chromatin modulators ATP-dependent nucleosome remodeling complexes SWI/SNF-A (BAF) and SWI/SNF-B (PBAF) have been shown to promote remodeling of the promoter regions of ISGs [182,183,184,185,186]. Furthermore, members of the histone acetyltransferases (HAT) family such as binding protein p300 [187], cAMP-responsive-element-binding protein (CREB)-binding protein (CBP) [188], and general control non-depressible 5 (GCN5) protein [189] mediate transcriptional activation through interactions with STAT1 and STAT2 directly on ISG promoters. While activation of these chromatin “remodelers” downstream type III IFN has currently not been investigated it is tempting to speculate that they will play a similar role like type I IFN treatment as STAT1 and STAT2 are critical regulators of type III dependent ISG induction. However, it may be that subtle differences in the kinetics or extent of action of the remodelers on chromatin might explain the differences observed in ISG kinetics expression between type I and III IFNs.

## 6. Conclusions

Over the past 15 years since the discovery of type III IFNs much work has been devoted to understanding whether they are redundant to type I IFNs. As more work emerges it becomes clear that while they have many similarities each IFN signaling cascade is finely tuned to allow for a unique antiviral environment. While more work is needed to characterize the type III IFN signaling cascade to a comparable level of type I IFNs, we already have a good view of this powerful new class of IFNs and their importance at mucosal surfaces. 

## Figures and Tables

**Figure 1 ijms-20-01445-f001:**
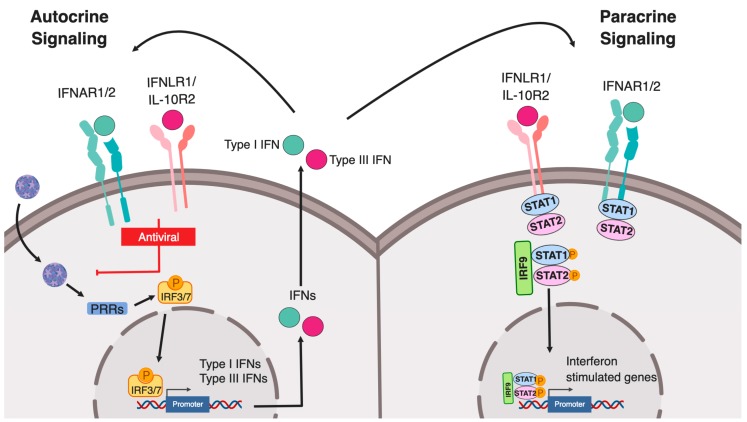
Type I and III interferon production following pathogen sensing. Schematic representing the canonical pathway of pathogen recognition receptor (PRR) sensing of intracellular pathogens. Pathogen associated molecular patterns (PAMPs) are recognized by PRRs leading to activation and phosphorylation of IRF3/7 which then translocate into the nucleus and drive the expression of both type I and III interferons. Both interferons are translated and secreted from the infected cell in an autocrine or paracrine manner. Upon binding of type I IFN to the IFNAR1/IFNAR2 and of type III IFN to the IFNLR1/IL-10R2 receptor, signal transductions are initiated leading to the formation of the ISGF3 complex (IRF9/p-STAT1/p-STAT2) which then acts as a transcription factor driving the expression of interferon stimulated genes.

**Figure 2 ijms-20-01445-f002:**
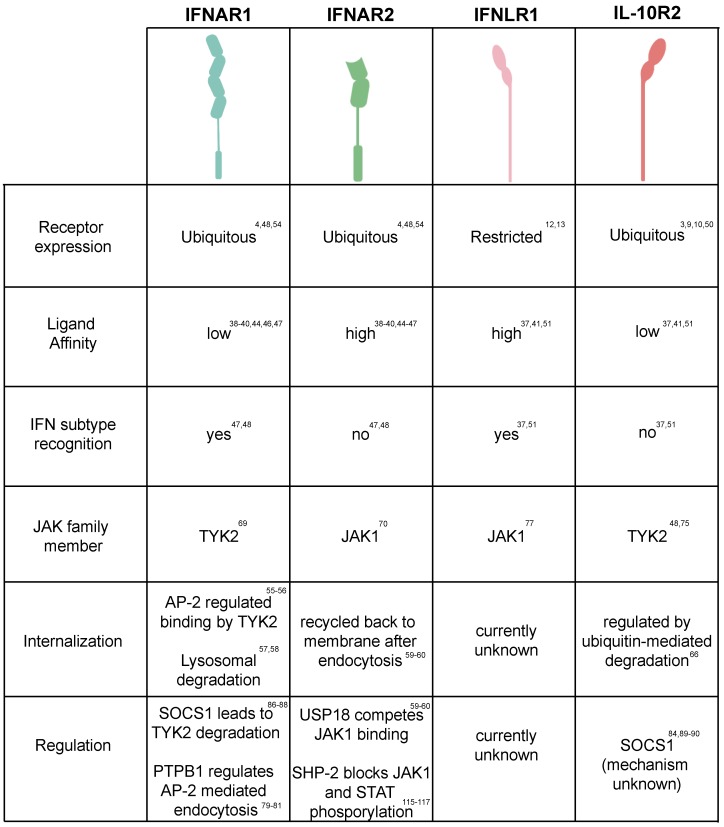
Regulation of interferon-mediated receptor signaling. Table summarizing the current state of the knowledge of both interferon receptor tropism, binding affinity for interferons, internalization/recycling mechanism and their known regulation.

**Figure 3 ijms-20-01445-f003:**
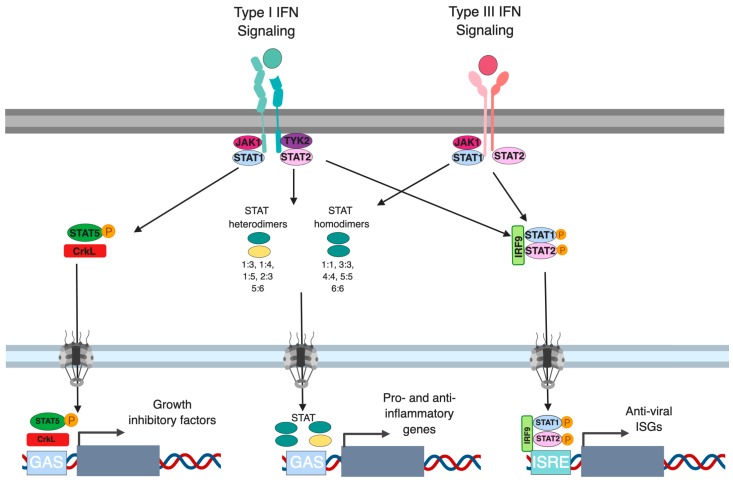
Signal transduction downstream type I and III interferon. Upon binding of type I IFN to the IFNAR1/IFNAR2 and of type III IFN to the IFNLR1/IL-10R2 receptor transactivation of JAK1 and TYK2 leads to the phosphorylation of STATs. Upon phosphorylation STAT molecules form three main complexes 1) STAT5/CrkL, 2) STAT homo- or heterodimers and 3) the ISGF3 complex (IRF9/p-STAT1/p-STAT2). All of these complexes act as transcription factors driving the production of interferon stimulated genes through the binding of specific motifs in their promoter sequences: GAS and ISRE elements.

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
