# Peer review of "Differential Regulation of Type I and Type III Interferon Signaling"

_ijms, 2019, doi:10.3390/ijms20061445_

Round 1
Reviewer 1 Report
This is a well-written review article that covers an important topic. It provides a great summary of recent developments in the field.
Criticism:
1. After printing, it was extremely difficult to read the figures. The authors should use larger fonts to describe the various elements shown in the illustrations.
2.Figure 2 could be improved by moving the GAS and ISRE motifs to the correct positions in the IFN-responsive genes. Such elements are most frequently found upstream of the transcription initiation site.
Author Response
Thank you for the very fast and positive review.
Point 1: We have updated the figures to increase the font size in both.
Point 2:We have update figure 2 showing the correct location of the GAS and ISRE elements. Thank you for catching this, it was a mistake in labeling on our part.
Reviewer 2 Report
This is a detailed and up-to-date review on innate antiviral interferons. It will be useful to the field. The figures are particularly nice and easy to understand. I have no major complaints about the content. However, there are a number of stylistic and spelling revisions to address prior to publication.
Major points:
Introduction: There is no mention of type II interferons. I realize that this review is focused on the more innate antiviral function of type I and type III IFNs, but IFNg should at least be mentioned and distinguished from the type I and type III IFNs.
Figure 1. I like the figure, it is simple and clear. However, the receptors should be labeled. I realize this is shown in Table 2 but it was not immediately clear looking at the figure.
Minor points:
Line 41. “homo sapiens” should be italicized and written as Homo sapiens
Line 51. The S in “IFNS” should be lower-case
Line 72. “janus” should be capitalized
Figure 1 legend. “Signals are transduced” would be a more clear way of expressing the events triggered by IFN/receptor binding than “signal transductions are initiated”
Lines 203-204. This sentence doesn’t make sense to me. Maybe it’s missing an “of” between “downstream” and “both”?
Line 207-208. This sentence also doesn’t make sense to me. Is “were” supposed to be “where”?
Line 251. “in-vitro” doesn’t need a hyphen and should be italicized (consistent with in vivo later in the same sentence)
Line 264. “It” should be lower-case
Line 485. Why are back-slashes used here?
Author Response
Thank you for the very rapid and positive review.
Major points:
We have now added the information on type II interferon to the introduction. Additionally, we have added the receptor labels to figure 1.
Minor points:
Thank you for taking the time to thoroughly read our manuscript and catching our small grammar errors. They have been corrected in the revised version as per your suggestion.